# Optimality of FSQ Tokens for Continuous Diffusion for Categorical Data with Application to Text-to-Speech

**Vadim Popov** [1 2] **Wenju Gu** [1] **Tasnima Sadekova** [1 2] **Georgii Aparin** [1 3] **Assel Yermekova** [1]

## Abstract

Continuous diffusion for categorical data is a framework belonging to the diffusion family and aiming at generating discrete data. The scientific interest to such models has been constantly increasing these days because researchers try to achieve a challenging goal of finding reasonable alternatives to autoregressive large language models. In this paper, we study the properties of the structure of the latent space corresponding to discrete tokens expressed in terms of Kullback-Leibler divergence on diffusion path measures and accuracy of the correct token prediction by the optimally trained diffusion model. We find that FSQ tokenization scheme has the latent space structure with the properties that make it best suited for continuous diffusion for categorical data as verified through rigorous theoretical analysis and numerical experiments. To validate our findings in real-life scenario, we train several text-to-speech diffusion models having speech tokens as intermediate acoustic features, and show that the one based on FSQ tokens indeed performs the best, and, moreover, it outperforms its strong LLM-based counterpart, at the same time being significantly smaller and faster.

## 1. Introduction

Diffusion probabilistic modeling (Ho et al., 2020; Song et al., 2021c) is a powerful framework initially developed for generating data belonging to continuous domains. It has demonstrated perfect results in many tasks such as image (Dhariwal & Nichol, 2021; Rombach et al., 2022; Gao et al., 2023), video (Luo et al., 2023), speech (Chen et al., 2021a;

Le et al., 2023) and music (Hawthorne et al., 2022; Novack et al., 2024) generation. Since the moment diffusion probabilistic models (DPMs) started to achieve state-of-the-art results in continuous domains, there have been numerous attempts to adapt this framework to discrete domains, most challenging of which is text. While the first such attempts (Hoogeboom et al., 2021) achieved modest results and had limited applicability to real-life scenarios, the latest findings made it possible to develop discrete diffusion models performing on par with some of Large Language Models (Brown et al., 2020), or LLMs for short, of moderate size in common NLP tasks (Gulrajani & Hashimoto, 2023; Lou et al., 2024; Nie et al., 2025) as well as in some specific areas like code generation (Fan et al., 2026).

The most successful diffusion models aiming at text generation typically have forward and reverse diffusion processes operating in discrete domains, and neural networks parameterizing reverse process in such models also predict probabilities of discrete entities (text tokens). For instance, in absorbing discrete DPM (Ou et al., 2025) forward and reverse diffusions correspond to, respectively, masking and unmasking a sequence of tokens, whereas neural network predicts fully unmasked sequence given its masked version. However, there exists a hybrid approach combining features of common discrete and continuous diffusion models. In such a hybrid framework called Continuous Diffusion for Categorical Data (CDCD), neural network predicts probabilities of discrete tokens, while both forward and reverse diffusion processes operate in continuous latent space containing embeddings of data tokens (Dieleman et al., 2022). The framework of CDCD models, however, has received relatively less attention compared to, for example, absorbing discrete diffusions, and one of the reasons is that the structure of the continuous latent space is typically optimized jointly with the diffusion model leading to unstable training (Dieleman et al., 2022; Gao et al., 2024; Nguyen et al., 2025) and, perhaps, suboptimal models. In this paper, we study properties of the structure of the latent space making it well-suited for CDCD models. We connect relative location of token embeddings with Kullback-Leibler (KL) divergence between measures associated with diffusion paths drifting towards these latent vectors. KL divergence between path measures is a meaningful metrics – some variant of it can

[1]Huawei Noah's Ark Lab [2]National Research University Higher School of Economics [3]National University of Science and Technology MISIS. Correspondence to: Vadim Popov <popov.vadim1@huawei.com>.

*Proceedings of the 43rd International Conference on Machine Learning*, Seoul, South Korea. PMLR 306, 2026. Copyright 2026 by the author(s).

be shown to be optimized during the process of training continuous DPMs (Song et al., 2021b).

Recently, there have been many efforts to obtain efficient discrete representation of objects belonging to modalities other than text. It can be explained by the increasing interest in the so-called omni-LLMs (Zhan et al., 2024; Xu et al., 2025), i.e. LLMs capable of handling multiple modalities (e.g. text, audio, images, video, etc.) in a unified way by means of tokenization. Thus, efficient discretization techniques, e.g. various types of Vector Quantization (VQ), are essential for omni-LLMs to work well. In our paper we demonstrate that a particular type of VQ, namely, Finite Scalar Quantization (Mentzer et al., 2024), or FSQ for short, leads to codebooks in the latent space with some optimality properties expressed in terms of KL divergence between diffusion path measures. Moreover, we set up the hypothesis that a CDCD model trained till optimality has the best prediction accuracy when the latent space structure is that of FSQ, prove this hypothesis in some particular cases and provide numerical experiments supporting it in the general case. Finally, we consider a realistic scenario of generating speech tokens by a CDCD model, and find out that, among several possible codebooks, the one used in FSQ in combination with the properly trained CDCD model indeed delivers the best quality of text-to-speech generation. Furthermore, the mentioned CDCD-based text-to-speech model outperforms its LLM-based counterpart in terms of speech intelligibility and zero-shot capabilities, and, notably, its CDCD backbone is around 10x smaller and more than 5x times faster than the LLM backbone of the competing algorithm.

Our main contributions are threefold: (i) for continuous diffusion processes, we theoretically establish the connection between forward diffusion parameters, distance between generated samples, and KL divergence between reverse diffusion paths leading to these samples; (ii) we formulate properties making FSQ codebooks best-suited for CDCD models and verify FSQ optimality both theoretically and through numerical experiments; (iii) we propose the novel CDCD-based text-to-speech model relying on FSQ speech tokens outperforming its LLM-based counterpart both in terms of synthesized speech quality and efficiency.

Our work has the following structure: we briefly overview related work in Section 2; all the necessary formulae and algorithms related to the CDCD framework, FSQ scheme and text-to-speech models are provided in Section 3; in Section 4 we describe our theoretical findings which we support by numerical experiments whose details can be found in Section 5 along with the description of the text-to-speech experiments. We conclude in Section 6. We will release training codes and the checkpoint of our best text-to-speech model at https://github.com/li1jkdaw/CDCD-TTS.

## 2. Related Work

As mentioned earlier, CDCD models are not as popular as vanilla continuous or discrete DPMs. As for FSQ method of data quantization, although it has found many applications, its properties with regard to diffusion modeling have not been extensively studied either. In this section we will briefly mention relevant work on the two mentioned topics. On the contrary, text-to-speech synthesis is a rapidly evolving field, so we review only the most relevant models either relying on speech tokens, or having diffusion models or flow matching (Lipman et al., 2023) as their backbone.

### 2.1. Diffusion Models for Categorical Data

With only a few exceptions (Nguyen et al., 2025), CDCD or similar models are trained for text generation. As for continuous modalities, it is more natural to use common continuous DPMs rather than discretize, say, images, and use discrete DPMs or CDCD models to generate discrete image tokens. CDCD (Dieleman et al., 2022) was one of the early attempts to combine features of discrete and continuous DPMs in a unified framework. It did it by keeping forward and reverse diffusions in continuous embedding space (we will refer to it as *latent* space) while parameterizing reverse process with the neural network predicting token probabilities and trained with cross-entropy loss. Later, researchers tried to improve upon this baseline by fixing instability issues or slightly modifying CDCD framework. Gao et al. (2024) suggest to add specific loss function to guarantee training stability, because optimizing pure diffusion loss in CDCD models may sometimes result in embedding collapse (note that in the original CDCD model token embeddings were learnable). Lovelace et al. (2023) and Liu et al. (2024) solve instability problem by relying on the pre-trained language models in building the latent space. Mahabadi et al. (2024) propose to use diffusions defined on vocabulary probability simplex instead of token embedding space. Shabalin et al. (2025) propose to build the latent space from token embeddings based on semantic similarity. To the best of our knowledge, our work is the first one to study the geometrical structure of the latent space for CDCD models in their original formulation.

### 2.2. FSQ method

Initially proposed for computer vision tasks (Mentzer et al., 2024) and used for image and video tokenization (Tang et al., 2024; Zhao et al., 2025), FSQ method allowing to use very low-dimensional latent spaces by locating token embeddings "uniformly" in a hypercube found more applications in speech. Its applications to this domain include not only conventional speech coding (Parker et al., 2025) where it helps to achieve extremely low bit-rates, but also zero-shot text-to-speech synthesis where LLM-based backbone gen-

erates FSQ speech tokens from input text autoregressively (Liao et al., 2024; Du et al., 2024; 2025). Furthermore, FSQ speech tokens can be used in speech LLMs (Ye et al., 2025) capable of various other tasks, e.g. automatic speech recognition, speech continuation, speech-to-speech question answering. etc.

### 2.3. Text-to-Speech

Modern text-to-speech (TTS) systems usually come with zero-shot capabilities (i.e. they can copy speaker's timbre, emotion and speaking style from a short reference audio) and typically consist of several modules one of them generating intermediate speech features (mel-spectrograms or speech tokens) from text by means of either LLM or diffusion/flow matching. As for the latter, one of the first hiqh-quality TTS models based on flow matching was VoiceBox (Le et al., 2023), further improved by getting rid of phoneme-level duration prediction in E2-TTS (Eskimez et al., 2024) and employing modern architectures in F5-TTS (Chen et al., 2024). While these models, as well as most of those relying on diffusions or flow matching, generated mel-spectrograms, there were developed a very few models (Ju et al., 2024) based on common discrete diffusions for generating speech tokens. In our paper, we start from one of the models generating intermediate speech tokens by means of LLM, namely CosyVoice2 (Du et al., 2024), and replace its backbone with CDCD, thus introducing the first CDCD-based TTS model.

## 3. Background

### 3.1. Continuous Diffusion for Categorical Data

Suppose we have a vocabulary of size $V$ with token ids $k = 1, .., V$ having corresponding $n$-dimensional embeddings $E = \{e_k\}_{k=1}^V$. Consider forward diffusion satisfying the following stochastic differential equation (SDE):

$$dX_t = f(t)X_t dt + g(t)dW_t, \ \ t \in [0, T], \qquad (1)$$

where $T$ is some finite positive time horizon, $W_t$ is Wiener process and functions $f(t)$ and $g(t)$ satisfy certain measurability conditions so that the SDE (1) has strong solution. In this paper we use Itô's calculus (Oksendal, 1992) when we manipulate with SDEs and corresponding stochastic processes. Diffusion time $t$ stands for the corruption level with $t = 0$ corresponding to uncorrupted latent vectors $X_0 \in E$ whereas the final level $t = T$ corresponds to the prior (standard normal $\mathcal{N}(0, \mathrm{I})$ in our experiments), i.e. we assume that $T$, $f(t)$ and $g(t)$ are such that $\mathrm{Law}(X_T) \approx \mathcal{N}(0, \mathrm{I})$ where I is $n \times n$ identity matrix. Note that in our theoretical analyses for the ease of exposition we always assume that our goal is to predict a single token given some conditioning $c$ rather than a sequence of tokens as it happens in most real-life scenarios.

It is a well-known fact (Kingma et al., 2021) that the SDE (1) allows for explicit solution

$$X_t = \alpha_t X_0 + \sigma_t \mathcal{N}(0, \mathrm{I}), \ \ t \in [0, T], \qquad (2)$$

where coefficients $\alpha_t$ and $\sigma_t$ are such that $\alpha_0 = 1, \sigma_0 = 0$, and they are connected with functions $f(t)$ and $g(t)$ by the following rule:

$$f(t) = \frac{\dot{\alpha}_t}{\alpha_t}, \ \ g^2(t) = 2\sigma_t \left( \dot{\sigma}_t - \frac{\dot{\alpha}_t}{\alpha_t}\sigma_t \right). \qquad (3)$$

CDCD model is trained to predict probabilities of tokens given their noisy latents $X_t$, diffusion time $t$ and conditioning $c$ by outputting logits and applying softmax function. So, neural network with parameters $\theta$ represents probability distribution $P_\theta(\cdot|X_t, t, c)$ and is trained by minimizing the following negative log-likelihood loss:

$$\mathcal{L}_{diff}(\theta) = \mathbb{E}_{(k, X_t) \sim p_{0,t}(\cdot, \cdot|c)}[-\log P_\theta(k|X_t, t, c)], \quad (4)$$

where diffusion time $t$ is sampled from uniform distribution $U[0, T]$ while ground truth label $k$ and corresponding noisy latent vector $X_t$ – from the joint distribution with the probability density function $p_{0,t}(\cdot, \cdot|c)$. In practice, we first sample ground truth label $k$ and then compute $X_t$ by the formula (2) where we replace $X_0$ with $k$-th token embedding $e_k$. By Bayes formula we can rewrite the above expression in terms of expectations with respect to the density function of the noisy latents $p_t(\cdot|c)$ and conditional probabilities $P_{0|t}(\cdot|X_t, c)$ of tokens given their noisy latent vectors:

$$\mathcal{L}_{diff}(\theta) = \mathbb{E}_{X_t \sim p_t(\cdot|c)} \mathbb{E}_{k \sim P_{0|t}(\cdot|X_t, c)}[-\log P_\theta(k|X_t, t, c)]$$

It follows from this formula that the parameters $\theta^*$ minimizing CDCD diffusion loss $\mathcal{L}_{diff}$ are such that

$$P_{\theta^*}(k|x, t, c) = P_{0|t}(k|x, c) \qquad (5)$$

for all $k = 1, .., V$, $t \in (0, T]$ and $x \in \mathrm{supp}\, p_t = \mathbb{R}^n$.

The above expression implies that once we have a well-trained neural network, we can estimate clean latent vector $X_0$ given its noisy version $X_t$:

$$X_\theta(X_t, t, c) = \sum_{k=1}^V P_\theta(k|X_t, t, c)e_k, \qquad (6)$$

and it follows from (5) that $X_{\theta^*}(X_t, t, c) = \mathbb{E}[X_0|X_t, c]$.

Generation with the trained CDCD model is possible in two different ways by solving either SDE or ordinary differential equation (ODE). In the remainder of the paper we will omit conditioning $c$ for brevity. In particular, denote probability density function of noisy latents $X_t$ by $p_t(\cdot)$ and write down the mentioned SDE and ODE:

$$d\hat{X}_t = (f(t)\hat{X}_t - g^2(t)\nabla \log p_t(\hat{X}_t))dt + g(t)d\hat{W}_t, \quad (7)$$

$$dX_t = (f(t)X_t - \frac{1}{2}g^2(t)\nabla \log p_t(X_t))dt, \qquad (8)$$

where $\hat{W}_t$ is a reverse-time Wiener process. Song et al. (2021c) showed that, given proper initial conditions, forward Kolmogorov equations on density functions of processes satisfying the SDE (1) and the ODE (8) are the same, and used the result obtained by Anderson (1982) to show that the SDE (7) describes reverse-time dynamics of the forward process $X_t$ satisfying the SDE (1). Thus, sampling discrete tokens is enabled by solving either the SDE (7) or the ODE (8) backwards in time from $t = T$ to $t = 0$ with the initial condition at $t = T$ being a random sample from $\mathcal{N}(0, \mathrm{I})$. If the resulting solution is $x_0$, then we generate token with id $\hat{k} = \arg\min_{k=1,..,V} \|x_0 - e_k\|_2^2$. The score function $\nabla \log p_t(x)$ is approximated by a function $s_\theta(x, t)$ expressed in terms of $X_\theta(x, t)$ by the formula

$$s_\theta(x, t) = -\frac{1}{\sigma_t^2} (x - \alpha_t X_\theta(x, t)). \qquad (9)$$

For the optimally trained neural network with parameters $\theta^*$ the above expression equals the true score function since

$$\nabla \log p_t(x) = -\frac{1}{\sigma_t^2} (x - \alpha_t \mathbb{E}[X_0 | X_t = x]), \qquad (10)$$

as known from the literature (e.g. see (Kingma et al., 2021)).

### 3.2. Finite Scalar Quantization

The Finite Scalar Quantization (Mentzer et al., 2024) algorithm contains the following encoding-decoding steps:

- Apply encoder *Enc* to input data $x$ and a bounded element-wise non-linearity $h$ after: $z = h(Enc(x))$.

- Round each of $n$ coordinates of the result to the nearest integer and apply decoder *Dec* to get reconstructed data $\hat{x}$: $\hat{x} = Dec(round(z))$

If outputs of the function $h : \mathbb{R}^n \to \mathbb{R}^n$ are bounded by $b \in \mathbb{Z}_+$ in absolute value, then each of $n$ coordinates can take up to $2b + 1$ values, resulting in the overall codebook size of $(2b + 1)^n$ tokens. In general, we can apply different bounded functions to different coordinates of the latent variable $z$, as well as employ asymmetric bounded non-linearities. The key feature of FSQ codebook is that its token embeddings (encoded data after rounding operation) are relatively low-dimensional vectors whose $i$-th coordinate can take any integer value between $a_i$ and $b_i$ where $i = 1, .., n$ and $a_i, b_i \in \mathbb{Z}$. It means that embeddings are located "uniformly" in the volume bounded by hyperplanes $z_i = a_i$ and $z_i = b_i$ for all $i = 1, .., n$.

In this paper, we consider two basic FSQ cases: $a_i = -1, b_i = 1$, and $a_i = 0, b_i = 1$ for all $i = 1, .., n$. We will refer to them as base3 and base2 FSQ respectively because of the number of different values each of $n$ coordinates of embeddings can take. For the base2 case, we linearly re-normalize FSQ latent space so that embedding coordinates are either $-1$ or $1$ (apply $z \mapsto 2z - 1$ after bounded non-linearity $h$ and replace rounding operation with sign function), because diffusion models usually operate on mean-normalized data. Thus, for base2 and base3 FSQ with dimension $n$ the codebook size is $2^n$ and $3^n$ respectively, and $L_\infty$ norm of each token embedding $e_k \in E$ is bounded by 1. Figure 1 (a-b) illustrates the codebooks $E$ of base2 and base3 FSQ schemes with dimension $n = 2$.

### 3.3. CosyVoice2

CosyVoice2 (Du et al., 2024) is a text-to-speech model consisting of three main modules: (i) text-to-token module represented by LLM initialized with the pre-trained Qwen2.5-0.5B (Yang et al., 2025a) trained to predict a sequence of speech tokens from an input text sequence; (ii) token-to-mel module based on flow matching; (iii) mel-to-wave module, i.e. vocoder. CosyVoice2 makes use of speech tokens obtained from encoder-decoder model with FSQ bottleneck trained on automatic speech recognition task rather than reconstruction. It means that the resulting FSQ tokens are such that it is easy to reconstruct linguistic content from them, but they are not guaranteed to contain other information such as speaker's timbre. Therefore, information about target speaker (CAM++ speaker embedding (Wang et al., 2023)) is fed to the second module converting a sequence of speech tokens to mel-spectrogram. As for the third module, CosyVoice2 uses HiFi-GAN vocoder (Kong et al., 2020) to convert mel-spectrogram to waveform.

CosyVoice2 is a TTS model capable of zero-shot voice cloning, i.e. it synthesizes speech with the same voice and in the same style as the reference speech given to it as input along with the text to synthesize. The first LLM-based module enables zero-shot mechanism by in-context learning as in VALLE-like TTS models (Zhang et al., 2023): before text we need to synthesize, we feed the text from the reference audio and the sequence of speech tokens corresponding to it to LLM so that it fits to the reference speaking style. As for conditioning the second flow matching module on reference audio, it is done similarly to VoiceBox (Le et al., 2023) where flow matching generates mel-spectrogram of the speech we need to synthesize and that of the reference speech jointly, which can be seen as mel-spectrogram "outpainting". One can refer to Figure 2 for illustration of CosyVoice2 general architecture and inference scheme.

## 4. Optimality Properties of FSQ Method

Throughout this section we assume that differential equations (1), (7) and (8), as well as their variants with the score

function replaced with its neural network approximation $s_\theta(x,t)$ defined in (9), have solutions on $[0,T]$, which requires certain measurability and Lipschitz assumptions on drift and diffusion coefficients. Moreover, following Anderson (1982), we assume that forward Kolmogorov equations on probability density functions of all the considered SDEs have unique smooth solutions. We also assume $\alpha_t > 0$ and $\sigma_t > 0$ for all $t \in (0, T]$.

### 4.1. Diffusion Path Measures

First let us establish the useful result regarding reverse-time diffusion bridges induced by the forward diffusion process $X_t$ defined by the SDE (1).

**Lemma 4.1.** *Consider $n$-dimensional stochastic process $X_t$ defined by the SDE (1) and condition it on some value $a \in \mathbb{R}^n$ at $t = 0$ and value $b \in \mathbb{R}^n$ at $t = T$. Denote the resulting diffusion bridge by $X_t^{a,b}$, i.e. $\mathrm{Law}(X_t^{a,b}) = \mathrm{Law}(X_t|X_0 = a, X_T = b)$ for $t \in [0,T]$. Then this diffusion bridge has reverse-time model $\hat{X}_t^{a,b}$ satisfying the following SDE:*

$$d\hat{X}_t^{a,b} = \left( \left( f(t) + \frac{g^2(t)}{\sigma_t^2} \right) \hat{X}_t^{a,b} - g^2(t)\frac{\alpha_t}{\sigma_t^2}a \right) dt \\ + g(t)d\hat{W}_t, \quad (11)$$

*for $t \in [0,T]$ with initial condition $\hat{X}_T^{a,b} = b$. Reverse-time $n$-dimensional Wiener process is denoted here by $\hat{W}_t$ and coefficients $\alpha_t$ and $\sigma_t$ are given in (3).*

Note that this lemma (as well as the statement we will formulate later in this section) does not rely on the fact that data distribution $\mathrm{Law}(X_0)$ is discrete, thus it can be useful not only when CDCD models are considered, but for common continuous DPMs as well. The proof of this lemma can be found in Appendix A and consists in applying Doob's transform (Rogers & Williams, 2000) to write down SDE for the diffusion bridge $X_t^{a,b}$ and Anderson's theorem (1982) to derive its reverse-time dynamics $\hat{X}_t^{a,b}$.

In practice, this lemma can be useful if we want to compare trajectories of the reverse diffusion $\hat{X}_t$ satisfying the SDE (7) starting in the same point $\hat{X}_T = b$ and finishing at time $t = 0$ in two different points $a_1$ and $a_2$, i.e. trajectories of the reverse-time diffusion bridges $\hat{X}_t^{a_1,b}$ and $\hat{X}_t^{a_2,b}$. In terms of generative modeling, when a diffusion model is trained well, it corresponds to comparing trajectories of this model starting from the same prior sample $b \sim \mathcal{N}(0, \mathrm{I})$ and generating two different data samples $a_1$ and $a_2$. The following statement formalizes this comparison by providing an expression in terms of KL divergence:

**Statement 4.2.** *Consider path measures defined by the distributions of continuous trajectories of the processes $\hat{X}_t^{a_1,b}$ and $\hat{X}_t^{a_2,b}$ for $t \in [\tau, T]$ for some $\tau \in (0, T)$ and denote*

them by $\mu_1$ and $\mu_2$ respectively. KL divergence between these measures is then calculated as

$$KL(\mu_1 || \mu_2) = \|a_1 - a_2\|_2^2 \cdot \frac{1}{2} \int_\tau^T \frac{\alpha_t^2}{\sigma_t^4} g^2(t) dt, \quad (12)$$

*if the above integral is finite.*

From the generative modeling perspective, Statement 4.2 establishes the connection between MSE distance between data samples $a_1$ and $a_2$ a well-trained diffusion model generates, functions $\alpha_t$, $\sigma_t$ and $g(t)$ describing the forward diffusion (1), and KL divergence between path measures defined by reverse diffusion trajectories finishing at data samples $a_1$ and $a_2$.

Statement 4.2 is a consequence of Girsanov theorem (Oksendal, 1992) and its proof can be found in Appendix B. KL divergence between path measures related to reverse diffusion is in general an important metrics describing how close certain reverse diffusion trajectories are to each other in distribution. For example, Song et al. (2021b) showed that training continuous DPMs by means of optimizing standard continuous diffusion MSE loss is equivalent to minimizing KL divergence between path measures corresponding to reverse diffusions with the true score function and its neural network approximation. So, we suggest that $KL(\mu_1 || \mu_2)$ has a connection with the difficulty of training a diffusion model capable of generating, in particular, two data samples $a_1$ and $a_2$. The intuition is that the smaller this KL divergence is, the harder it is to train a model distinguishing well between generative trajectories leading to $a_1$ and $a_2$.

As far as FSQ latent space structure is concerned, it has some optimal properties in terms of relative position of token embeddings $e_k$, which we now can connect with KL divergence between reverse diffusion path measures and, hence, with the difficulty of CDCD training. Consider the following metrics $D(E)$ defined for a codebook $E = \{e_k\}_{k=1}^V$:

$$D(E) := \frac{1}{V} \sum_{k=1}^V \min_{i \neq k} \|e_i - e_k\|_2^2. \quad (13)$$

One can see from the above definition that this is the *average nearest neighbour distance*. Statement 4.2 allows us to interpret maximizing this metrics as being conservative about how close in distribution diffusion trajectories corresponding to different tokens should be and trying to help neural network to train better in the "worst-case" scenario. The following theorem establishes local optimality of FSQ codebooks in terms of average nearest neighbour distance.

**Theorem 4.3.** *In terms of average nearest neighbour distance $D$, the codebooks $E_{FSQ} = \{e_k\}_{k=1}^V$ of base2 and base3 FSQ methods are locally optimal in the class of all codebooks with $V$ entries such that $\|e_k\|_\infty \leq 1$ for all $k = 1, .., V$, i.e. any sufficiently small perturbation of a vector from the codebook $E_{FSQ}$ leads to decreasing $D$.*

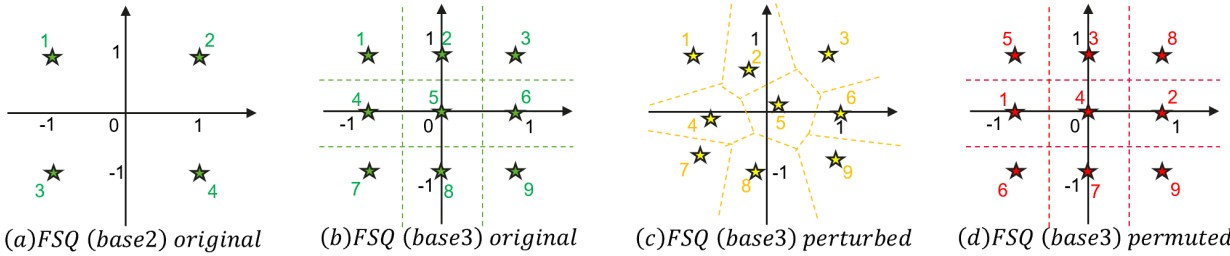

*Figure 1.* 2-dimensional FSQ latent space structure. All points lie inside the square $|y_1| \leq 1, |y_2| \leq 1$. Colored numbers stand for token ids. Dotted lines are the bounds of areas $\Omega_k^Y$ for token ids $k = 1, .., b^2$ where $b$ equals to the base, i.e. 2 for (a) and 3 for (b-d).

Whereas this theorem (see Appendix C for its proof) establishes local optimality of FSQ codebooks, Section 5.1 contains numerical experiments supporting the hypothesis about their global optimality.

### 4.2. Prediction Accuracy

In Section 4.1 we described properties of CDCD latent space that potentially can help this model train better. In this section, we assume that the CDCD model has already been trained till optimality and discuss how the latent space structure affects token prediction accuracy.

Denote token prior probabilities by $p_k$ for $k = 1, .., V$. Recall that the probability distribution parameterized with the optimally trained CDCD equals $P_{0|t}(k|x)$ (5). By (2) and Bayes formula we have

$$P_{0|t}(k|x) \propto p_k \exp\left(-\frac{1}{2\sigma_t^2}\|x - \alpha_t e_k\|_2^2\right). \quad (14)$$

Denote the area of latent space where we predict token id $k$ at diffusion time $t$ by $\Omega_k^X(t)$, i.e.

$$\Omega_k^X(t) = \{x \in \mathbb{R}^n : k = \arg\max_j P_{0|t}(j|x)\}. \quad (15)$$

With this notation, we can introduce the *average prediction accuracy* metrics $A(E, t)$ defined by

$$A(E, t) = \sum_{k=1}^{V} P(X_0 = e_k, X_t \in \Omega_k^X(t)). \quad (16)$$

Our hypothesis is that for every $t \in [0, T]$ $A(E, t)$ is maximized in the class of all codebooks of size $V$ with entries bounded by 1 in $L_\infty$ norm for FSQ codebooks $E_{FSQ}$ given that prior token probabilities are equal, i.e. $p_k = 1/V$ for all $k = 1, .., V$.

We refer to this hypothesis as *Best Accuracy Hypothesis* and for 1-dimensional latent space we prove it in Appendix D while for higher dimensions we provide numerical evidence supporting it in the experiments described in Section 5.1. If this hypothesis is true indeed, it means that well-trained

CDCD models generating FSQ tokens better predict token probabilities in terms of top-1 accuracy. Although it does not necessarily imply better generation quality because, as one can see from the formulae (6) and (9), sampling from CDCD involves computing probabilities of *all* tokens rather than just the most probable one, it can still be considered a meaningful metrics since it has a clear interpretation.

In Appendix D we show that it is possible to consider stochastic process $Y_t = X_t/\alpha_t$ whose corresponding latent space areas $\Omega_k^Y(t)$ defined similarly to (15) do not actually depend on $t$ when all $p_k$ are equal. Figures 1(b) and 1(c) illustrate these areas for original base3 FSQ codebook and its slightly perturbed version.

The main limitation of the argument in this section is that we consider the case with equal prior token probabilities which is not always true in practice. The opposite case requires different treatment and corrections compensating for the imbalance, and we leave it as future research direction.

## 5. Experiments

### 5.1. Numerical Experiments

First, we performed experiments devoted to global optimality of FSQ codebooks in terms of square root of average nearest neighbour distance. As follows from the structure of FSQ latent space, for base2 and base3 FSQ codebooks we have these metrics equal to 2 and 1 respectively. We randomly generated 10000 codebooks with entries bounded in absolute value by 1 in $L_\infty$ norm for base2 case for dimensionalities 2, 3, 4 and 8 and achieved the highest values of this metrics 1.993, 1.984, 1.941 and 1.818 which *all* are less than 2. For base3 case we obtained the values 0.969, 0.913, 0.871 and 0.825 for the same dimensionalities which *all* are less than 1. These experiments show that average nearest neighbour distance at least for *most* of possible codebooks is less than that for FSQ codebooks which supports the hypothesis about the global optimality of FSQ codebooks whose local optimality in terms of average nearest neighbour distance is established in Theorem 4.3.

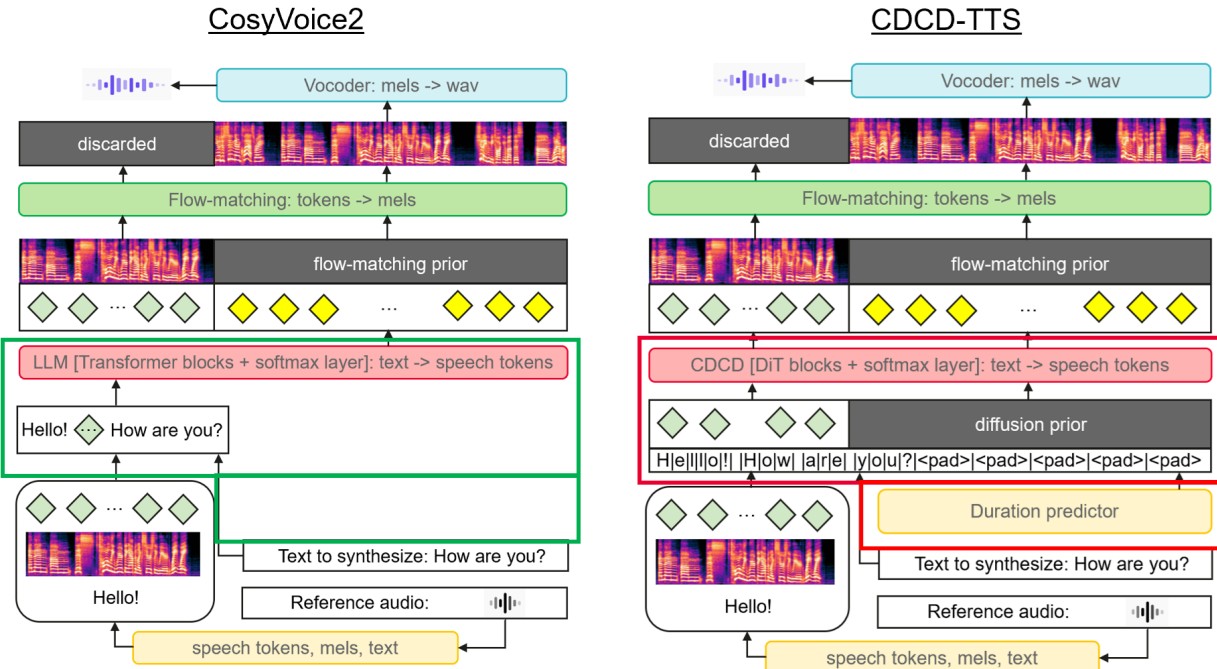

*Figure 2.* Comparison of CosyVoice2 (left) and CDCD-TTS (right). Green and red boxes show the difference between two models. Green diamonds stand for reference speech tokens, yellow – for those generated by a model. The gray "discarded" box means that once we generated corresponding mel features, we no longer use them (they are just necessary to condition the flow-matching module during its inference).

Next, we tried to verify Best Accuracy Hypothesis. We considered 1000 different codebooks for base2 and base3 cases and dimensionalities 2, 3 and 4. For each codebook, we performed Monte-Carlo sampling with around 10M samples to estimate accuracy. For base2 case FSQ accuracies for the mentioned dimensionalities were 38.179%, 23.580% and 14.573%, while the best accuracies among 1000 generated codebooks were 37.389%, 22.816% and 13.789%. For base3 case FSQ accuracies are 17.052%, 7.044% and 2.889% with the best accuracies among generated codebooks being 16.659%, 6.739% and 2.731%. These numerical experiments support Best Accuracy Hypothesis for dimensionalities larger than one.

Finally, we conducted experiments on unconditional toy data generation to verify that the geometrical properties of FSQ method give benefits when we train CDCD models. We considered 2-dimensional latent space and 4-element sequences where each element belonged to the vocabulary of size 4 (for base2 case) or 9 (base3 case). We trained 5 different CDCD models with 5 different random seeds for each case for the latent space of FSQ method and for the latent space with slightly perturbed token embeddings till convergence and evaluated average of log-KL divergence between ground truth distribution (different sequences had different prior probabilities) and the distribution of the gen-

erated sequences (with 100 denoising steps). We found that models trained with the original FSQ tokens gave better log-KL than those trained with the perturbed token embeddings both in base2 and base3 cases: $-9.55$ vs $-8.68$, and $-7.16$ vs $-6.72$ correspondingly. Neural networks considered in this experiment were a stack of 4 layers of convolutions with kernel size 3 and 256 channels with GELU non-linearity and sinusoidal positional embedding to represent diffusion time $t$. For this toy example we verified that the latent space geometry of FSQ scheme gives advantage to CDCD models by itself; in the next section, we will check what happens if we incorporate CDCD operating on FSQ tokens into a more complicated architecture.

### 5.2. Text-to-Speech Experiments

We have derived results suggesting optimality of FSQ codebooks for CDCD models and verified several of them in numerical experiments on toy data, and in this section we apply CDCD for token generation in a real-life scenario.

We started from CosyVoice2 TTS model utilizing base3 FSQ speech tokens (see Section 3.3) with 8-dimensional embeddings (resulting in the total of $3^8 = 6561$ speech tokens sampled at 25Hz) and essentially replaced its LLM-based text-to-token module with CDCD while keeping the

remaining modules as well as FSQ speech tokenizer unchanged. Because of non-autoregressive nature of generation with CDCD model, we had to additionally introduce a statistical duration prediction module predicting a single number – how many speech tokens should be generated given an input text. This module does not rely on neural networks and was chosen for its simplicity because in our preliminary experiments it did not lead to quality degradation compared to more complicated duration predictors, e.g. the one borrowed from F5-TTS based on a neural network. More details can be found in Appendix E.

We borrowed model architecture of CDCD from the corresponding F5-TTS (Chen et al., 2024) module generating continuous acoustic features with continuous DPM, and just added softmax layer on top of DiT (Peebles & Xie, 2023) to enable token ids prediction as required by the CDCD framework. We also used the same approach of conditioning CDCD on reference text and reference speech tokens – FSQ embeddings of reference tokens were concatenated to the main neural network input, i.e. to noisy latent vectors, and, similarly, reference text was concatenated to the text TTS model had to generate, and right padding was added to fit speech tokens sequence length. Both CosyVoice2[1] (together with FSQ speech tokenizer) and F5-TTS[2] have official GitHub implementation that we extensively utilized while implementing our TTS model we called CDCD-TTS. Figure 2 provides comparison of CosyVoice2 and CDCD-TTS, and Appendix E contains a detailed description of the architecture of the latter.

We trained four CDCD-based TTS models on $65k$ hours of English speech data from LibriLight (Kahn et al., 2020), GigaSpeech (Yang et al., 2025b) and English subset of Emilia dataset (He et al., 2024). *FSQ-original* stands for the original model we call CDCD-TTS trained to generate true FSQ speech tokens. *FSQ-perturb* stands for the model where FSQ token embeddings were slightly perturbed ($\pm 1$ coordinates remained unperturbed while each 0 coordinate was randomly shifted either to $+0.5$ or to $-0.5$ so that the convex hull is still the same hypercube and the overall data scale remains unchanged). *FSQ-permute* stands for the same token embeddings as in the original FSQ scheme but with permuted token ids. These three models had exactly the same architectures and were trained for 1.5M steps with warmup with the maximum learning rate $10^{-4}$ and batch size of 600 seconds on 8 V100 GPUs. The last CDCD-based model denoted by *RVQ* was trained to predict speech tokens coming from an alternative hierarchical quantization scheme called Residual Vector Quantization (Vasuki & Vanathi, 2006) employed in EnCodec (Défossez et al., 2023) speech tokenizer having 8 VQ layers. Since EnCodec

[1] https://github.com/FunAudioLLM/CosyVoice
[2] https://github.com/SWivid/F5-TTS

*Table 1.* Evaluation of TTS models on SEED *test-en* set. F5-TTS and CosyVoice3 results in italics are given as a reference.

|  | WER | SIM | MOS | EMO |
|---|---|---|---|---|
| *RVQ-25* | 21.3% | 0.382 | 2.932 | 52.0% |
| *FSQ-permute-25* | 15.4% | 0.588 | 3.631 | 70.1% |
| *FSQ-original-5* | 2.39% | **0.654** | 4.093 | 71.7% |
| *FSQ-perturb-5* | 3.10% | 0.647 | 3.834 | 70.6% |
| *FSQ-original-8* | 2.10% | **0.654** | **4.119** | 72.2% |
| *FSQ-perturb-8* | 2.32% | 0.649 | 4.030 | 71.8% |
| *FSQ-original-12* | 2.05% | 0.653 | **4.120** | 72.3% |
| *FSQ-perturb-12* | 2.14% | 0.647 | 4.088 | 72.1% |
| *FSQ-original-25* | **2.00%** | 0.653 | 4.119 | **72.7%** |
| *FSQ-perturb-25* | 2.03% | 0.648 | **4.118** | 72.3% |
| *CosyVoice2(2024)* | 2.57% | **0.652** | 4.077 | 72.2% |
| *F5-TTS(2024)* | 1.83% | 0.665 | 3.754 | 71.4% |
| *CosyVoice3(2025)* | 1.68% | 0.695 | 3.937 | 72.7% |

vectors have higher dimensionality (namely, 128), we had to reduce batch size to 300 seconds and maximum learning rate to $5 \cdot 10^{-5}$. *RVQ* model generates only the first RVQ layer of tokens while the remaining 7 layers are generated by the non-autoregressive module borrowed from VALLE-X TTS model (Zhang et al., 2023).

By choosing to train *FSQ-original* text-to-speech model, we consider the most popular VQ technique with the fixed codebook used for obtaining speech tokens. Note that another popular VQ technique LFQ (Yu et al., 2024) is the same as base2 FSQ from the point of view of relative position of token embeddings. As for VQ techniques with codebooks changing during VQ training, we choose *RVQ* as a baseline common for speech applications and *FSQ-perturb* is meant to account for various VQ techinques with the single codebook (e.g. VQ-VAE) where trained embeddings may have almost arbitrary relative positions. For fair comparison between CDCD-based models, we used the same backbone architecture having approximately 45M parameters with the only difference that we accounted for different input noisy embedding size (8 for FSQ vs 128 for RVQ). Also, for the same purpose training conditions and hyperparameters were chosen as similar as possible.

At generation stage for CDCD models we use different number of denoising steps belonging to $\{5, 8, 12, 25\}$ (this number is given as a postfix to the model names in Table 1) and employ DDIM solver (Song et al., 2021a) to solve the ODE (8). We use classifier-free guidance related to text conditioning with weight 0.5 and start generation from random samples belonging to $\mathcal{N}(0, (2/3)^2 I)$ rather than $\mathcal{N}(0, I)$ as it gives significantly better results in terms of intelligibil-

ity and sound quality on our development set. All models are tested on SEED-TTS (Anastassiou et al., 2024) *test-en* test set containing roughly 1000 sentences. We synthesize each of them 4 times with each model under comparison and report metrics aggregated by means of median function. Speech intelligibility is measured with Word Error Rate (WER) and speaker similarity with the target speaker (SIM) – as cosine similarity between WavLM (Chen et al., 2021b) speaker embeddings tuned for speaker verification task. These two metrics are measured with officially provided tool[3]. As for the overall speech quality, we measure it with UTMOS[4], proxy to human 5-point Mean Opinion Scores (MOS). Finally, we test how well TTS models copy emotions from reference speech by running *emotion2vec* model[5] and checking whether emotion labels assigned by this model to reference and synthesized speech are the same. The corresponding EMO score is just the accuracy of these predictions. We compare our CDCD-based TTS models with their LLM-based counterpart CosyVoice2 and also add a popular contemporary model F5-TTS (Chen et al., 2024) and one of the current state-of-the-art zero-shot TTS models CosyVoice3 (Du et al., 2025) to the comparison as references. The results of this comparison are given in Table 1.

This table demonstrates that *FSQ-original* model consistently outperforms second best *FSQ-perturb* model for any number of reverse diffusion steps, although the difference becomes marginal when the number of steps is large enough. It implicitly supports local optimality of FSQ codebooks for the practical case of imperfectly trained CDCD models with limited capacity (note that given optimally trained network and "infinite" inference denoising steps, for any reasonable latent space geometry we should have perfect data reconstruction). On the other hand, *FSQ-permute* and *RVQ* perform not very well even when rather large number of steps are used at inference, which suggests that FSQ tokenization may be globally optimal for the CDCD framework, but, at the same time, it is not sufficient just to randomly assign tokens to FSQ embeddings – token ids and FSQ embeddings should be mapped to each other as a result of autoencoder training. Also, *RVQ* model performs worse than all FSQ counterparts, and it can be perhaps explained by the fact that for the model with the same capacity it is difficult to account for embeddings with significantly larger embedding size (128 vs 8).

Remarkably, our best CDCD-TTS model outperforms its LLM-based counterpart by a large margin in speech intelligibility and speech naturalness as demonstrated in Table 1. Moreover, its DiT backbone is 10x smaller than 0.5B LLM

---

[3] https://github.com/BytedanceSpeech/seed-tts-eval
[4] https://huggingface.co/spaces/sarulab-speech/UTMOS-demo/tree/main
[5] https://huggingface.co/emotion2vec

in CosyVoice2 and, depending on duration of reference speech utterance, is $5 - 10x$ faster, allowing the whole CDCD-TTS algorithm to achieve real-time factor of around $0.2 - 0.3$, i.e. it synthesizes speech $3 - 5x$ faster than real time on a GPU with only 25 reverse diffusion steps.

## 6. Conclusion

In this paper we studied continuous diffusion for categorical data and the latent space properties allowing to train such models with good quality. We established the connection between reverse diffusion path measures and the distance between data samples generated along these trajectories, and used this connection to interpret a geometrical property of FSQ codebooks in terms of KL divergence between trajectories leading to token embeddings and their suitability for CDCD training. Also, we set up the hypothesis about average prediction accuracy being maximized for FSQ latent space in case of optimally trained diffusions, and provided some theoretical and empirical evidence for that. Finally, we trained the first CDCD-based text-to-speech model capable of efficient zero-shot voice cloning of very high quality. Its superior performance compared to LLM-based counterpart suggests that CDCD models are a promising research direction.

## Acknowledgements

The work of Vadim Popov and Tasnima Sadekova was supported by the grant for research centers in the field of AI provided by the Ministry of Economic Development of the Russian Federation in accordance with the agreement **000000C313925P4E0002** and the agreement with HSE University №139-15-2025-009.

## Impact Statement

This paper's topic is generative deep learning which has potential risks when used improperly or with bad intentions. Theoretical part of the paper does not seem to lead to any specific societal consequences. As for the text-to-speech model we introduce, it is capable of zero-shot voice cloning, thus it must be used with care to protect privacy as we will definitely mention when we open-source it.

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

# A. Proof of Lemma 4.1.

*Proof.* Consider the forward diffusion process $X_t$ satisfying the SDE (1):

$$dX_t = f(t)X_t dt + g(t)dW_t, \ \ t \in [0, T].$$

Denote its conditional probability density function by $p_{t|s}(\cdot|\cdot)$. From the literature on diffusion generative modeling (Kingma et al., 2021) we know that the generalized version of the formula (2) describing conditional distribution with the density $p_{t|s}$ holds:

$$\text{Law}\,(X_t|X_s) = \mathcal{N}(\alpha_{t|s}X_s, \sigma_{t|s}^2\,\text{I}), \ \ \alpha_{t|s} = \frac{\alpha_t}{\alpha_s}, \ \ \sigma_{t|s}^2 = \sigma_t^2 - \alpha_{t|s}^2\sigma_s^2, \ \ 0 \le s < t \le T. \tag{17}$$

In order to condition the forward diffusion $X_t$ on its endpoints to obtain the process $X_t^{a,b}$ such that $\text{Law}\,(X_t^{a,b}) = \text{Law}\,(X_t|X_0 = a, X_T = b)$, we can use Doob's $h$-transform:

$$h^{a,b}(x,t) := \frac{p_{T|t}(b|x)}{p_{T|0}(b|a)},$$

and write down the following SDE (Rogers & Williams, 2000; Zhou et al., 2024):

$$dX_t^{a,b} = (f(t)X_t^{a,b} + g^2(t)\nabla_x \log h^{a,b}(X_t^{a,b}, t))dt + g(t)dW_t, \ \ t \in [0, T].$$

Formula (17) implies that $\nabla_x \log h^{a,b}(x,t) = \nabla_x \log p_{T|t}(b|x) = \alpha_{T|t}(b - \alpha_{T|t}x)/\sigma_{T|t}^2$, thus $X_t^{a,b}$ satisfies the SDE

$$dX_t^{a,b} = \left(\alpha_{T|t}\frac{g^2(t)}{\sigma_{T|t}^2}b + \left(f(t) - \alpha_{T|t}^2\frac{g^2(t)}{\sigma_{T|t}^2}\right)X_t^{a,b}\right)dt + g(t)dW_t, \ \ t \in [0, T], \tag{18}$$

with the initial condition $X_0^{a,b} = a$.

Now let us find its time-reverse model by applying Anderson's theorem (1982). To do this, we need to know marginal probability densities of the diffusion bridge process $X_t^{a,b}$. It is known that these densities are Gaussian (Kingma et al., 2021):

$$\text{Law}\,(X_t^{a,b}) = \text{Law}\,(X_t|X_T = b, X_0 = a) = \mathcal{N}\left(\alpha_t\frac{\sigma_{T|t}^2}{\sigma_T^2}a + \alpha_{T|t}\frac{\sigma_t^2}{\sigma_T^2}b, \frac{\sigma_t^2\sigma_{T|t}^2}{\sigma_T^2}\,\text{I}\right),$$

so the score function corresponding to the diffusion bridge $X_t^{a,b}$ can be expressed as

$$\nabla_x \log p_t^{a,b}(x) = -\frac{\sigma_T^2}{\sigma_t^2\sigma_{T|t}^2}\left(x - \alpha_t\frac{\sigma_{T|t}^2}{\sigma_T^2}a - \alpha_{T|t}\frac{\sigma_t^2}{\sigma_T^2}b\right) = -\frac{\sigma_T^2}{\sigma_t^2\sigma_{T|t}^2}x + \frac{\alpha_t}{\sigma_t^2}a + \frac{\alpha_{T|t}}{\sigma_{T|t}^2}b. \tag{19}$$

Anderson's theorem implies that reverse-time model $\hat{X}_t^{a,b}$ of the diffusion bridge $X_t^{a,b}$ satisfies the SDE

$$d\hat{X}_t^{a,b} = \left(\alpha_{T|t}\frac{g^2(t)}{\sigma_{T|t}^2}b + \left(f(t) - \alpha_{T|t}^2\frac{g^2(t)}{\sigma_{T|t}^2}\right)\hat{X}_t^{a,b} - g^2(t)\nabla \log p_t^{a,b}(\hat{X}_t^{a,b})\right)dt + g(t)d\hat{W}_t,$$

where $\hat{W}_t$ is reverse-time Brownian motion and the initial condition is $\hat{X}_T^{a,b} = b$. Plugging in the expression for the score function from (19) into the above SDE and using the identity $\sigma_{T|t}^2 = \sigma_T^2 - \alpha_{T|t}^2\sigma_t^2$ following from (17), we finally obtain the desired SDE:

$$d\hat{X}_t^{a,b} = \left(\left(f(t) + \frac{g^2(t)}{\sigma_t^2}\right)\hat{X}_t^{a,b} - g^2(t)\frac{\alpha_t}{\sigma_t^2}a\right)dt + g(t)d\hat{W}_t, \ \ t \in [0, T], \tag{20}$$

with the initial condition $\hat{X}_T^{a,b} = b$. This SDE is to be solved backwards in time. $\qquad\square$

# B. Proof of Statement 4.2.

*Proof.* Consider two different reverse diffusion bridges $\hat{X}_t^{a,b}$ for $a = a_1$ and $a = a_2$ with time $t$ running backwards from $t = T$ to $t = 0$. Define random processes $Z_t^i := \hat{X}_{T-t}^{a_i,b}$ for $i = 1, 2$. They are now forward-time stochastic processes satisfying SDEs:

$$dZ_t^i = -\left(\left(f(T-t) + \frac{g^2(T-t)}{\sigma_{T-t}^2}\right)Z_t^i - g^2(T-t)\frac{\alpha_{T-t}}{\sigma_{T-t}^2}a_i\right)dt + g(T-t)dW_t, \tag{21}$$

where $i = 1, 2$, $W_t$ is a forward-time Brownian motion, and time $t$ increases from $t = 0$ to $t = T$. Diffusions $Z_t^i$ are two processes with the same diffusion coefficient and starting at the same point $Z_0^i = X_T^{a_i,b} = b$. It suggests that we can now apply Girsanov theorem.

Consider process $\widetilde{W}_t$ such that

$$d\widetilde{W}_t = dW_t + g(T-t)\frac{\alpha_{T-t}}{\sigma_{T-t}^2}(a_1 - a_2)dt.$$

Denote probability measure corresponding to the probability space we consider by $\mathbb{P}$. In particular, $\{W_t\}_{t\in[0,T]}$ is a Brownian motion with respect to $\mathbb{P}$. Let $\tau$ be any positive number in $(0, T]$. The Girsanov theorem I from (Oksendal, 1992) implies that, if Novikov's condition is satisfied, i.e. if

$$\mathbb{E}\left[\exp\left(\|a_1 - a_2\|_2^2 \cdot \frac{1}{2}\int_0^{T-\tau} g^2(T-t)\frac{\alpha_{T-t}^2}{\sigma_{T-t}^4}dt\right)\right] < \infty \iff \frac{1}{2}\int_\tau^T g^2(t)\frac{\alpha_t^2}{\sigma_t^4}dt < \infty,$$

then $\{\widetilde{W}_t\}_{t\in[0,T-\tau]}$ is a Brownian motion with respect to probability measure $\mathbb{Q}$ defined by its Radon-Nikodym derivative

$$\frac{d\mathbb{Q}}{d\mathbb{P}} = \exp\left(-\int_0^{T-\tau} g(T-t)\frac{\alpha_{T-t}}{\sigma_{T-t}^2}(a_1 - a_2)dW_t - \frac{1}{2}\int_0^{T-\tau}\|a_1 - a_2\|_2^2 g^2(T-t)\frac{\alpha_{T-t}^2}{\sigma_{T-t}^4}dt\right). \tag{22}$$

We can rewrite the SDE for the process $Z_t^1$ in the following way:

$$dZ_t^1 = -\left(\left(f(T-t) + \frac{g^2(T-t)}{\sigma_{T-t}^2}\right)Z_t^1 - g^2(T-t)\frac{\alpha_{T-t}}{\sigma_{T-t}^2}a_2\right)dt + g(T-t)d\widetilde{W}_t.$$

Comparing this SDE to the one for $Z_t^2$

$$dZ_t^2 = -\left(\left(f(T-t) + \frac{g^2(T-t)}{\sigma_{T-t}^2}\right)Z_t^2 - g^2(T-t)\frac{\alpha_{T-t}}{\sigma_{T-t}^2}a_2\right)dt + g(T-t)dW_t.$$

we can deduce that the process $\{Z_t^1\}_{t\in[0,T-\tau]}$ has the same distribution under the probability measure $\mathbb{Q}$ as the process $\{Z_t^2\}_{t\in[0,T-\tau]}$ under the probability measure $\mathbb{P}$.

Now consider diffusion path measures defined by distributions of the trajectories $\{\hat{X}_t^{a_1,b}\}_{t\in[\tau,T]}$ and $\{\hat{X}_t^{a_2,b}\}_{t\in[\tau,T]}$. Denote them $\mu_1$ and $\mu_2$ respectively. For any measurable set $C$ of continuous trajectories in $\mathbb{R}^n \times [\tau, T]$ we have

$$\mu_1(C) = \mathbb{P}(\{\hat{X}_t^{a_1,b}\}_{t\in[\tau,T]} \in C) = \mathbb{P}(\{Z_t^1\}_{t\in[0,T-\tau]} \in C)$$

$$\mu_2(C) = \mathbb{P}(\{\hat{X}_t^{a_2,b}\}_{t\in[\tau,T]} \in C) = \mathbb{P}(\{Z_t^2\}_{t\in[0,T-\tau]} \in C) = \mathbb{Q}(\{Z_t^1\}_{t\in[0,T-\tau]} \in C)$$

Compute KL divergence between $\mu_1$ and $\mu_2$ taking into account that Itô's integral of a deterministic square-integrable function is a normal random variable with zero mean:

$$KL(\mu_1\|\mu_2) = \mathbb{E}_{\mu_1}\left[-\log\frac{d\mu_2}{d\mu_1}\right] = \mathbb{E}\left[-\log\frac{d\mathbb{Q}}{d\mathbb{P}}\right] = \mathbb{E}\left[\int_0^{T-\tau} g(T-t)\frac{\alpha_{T-t}}{\sigma_{T-t}^2}(a_1 - a_2)dW_t\right]$$

$$+ \frac{1}{2}\mathbb{E}\left[\int_0^{T-\tau}\|a_1 - a_2\|_2^2 g^2(T-t)\frac{\alpha_{T-t}^2}{\sigma_{T-t}^4}dt\right] = \|a_1 - a_2\|_2^2 \cdot \frac{1}{2}\int_\tau^T g^2(t)\frac{\alpha_t^2}{\sigma_t^4}dt.$$

$\square$

# C. Proof of Theorem 4.3

*Proof.* Consider the first base2 case. Suppose we modified the token embedding $e$ having $m$ coordinates equal to $-1$ and $p$ coordinates $+1$, where $p + m$ is the latent space dimensionality and $p \geq 0, m \geq 0$. Suppose the modified embedding $e'$ has coordinates $-1 + \delta_i$ and $+1 - \beta_j$ instead of $-1$ and $+1$ for $i = 1, .., p$ and $j = 1, .., m$. Perturbation values $\delta_i$ and $\beta_j$ are non-negative because we need to have $\|e'\|_\infty \leq 1$. Denote the largest absolute value of per-coordinate perturbations by $\gamma$, i.e. $\gamma := \max\left(\max_{i=1,..,p} |\delta_i|, \max_{j=1,..,m} |\beta_j|\right)$. The overall perturbation $\Delta = \sum_{i=1}^p \delta_i^2 + \sum_{j=1}^m \beta_j^2$ is assumed to be sufficiently small, i.e. $\Delta = \bar{o}(\gamma)$. Perturbing a single codebook entry $e$ by modifying it to $e'$ could not increase nearest neighbour distance for any other embedding, since they all have at least two neighbours on the distance 2 (except for the trivial 1-dimensional case for which the statement of the theorem is obvious) which is the nearest neighbour distance for all token embeddings in base2 FSQ scheme. Now let us prove that the nearest neighbour distance decreases for the modified $e'$.

Suppose that the largest perturbation with magnitude $\gamma$ corresponds to $-1$ coordinate modified to $-1 + \gamma$. Consider embedding $e^*$ differing from $e$ only in this coordinate (having $+1$ there instead of $-1$). Squared distance between $e'$ and $e^*$ is then calculated as $\|e' - e^*\|_2^2 = \Delta - \gamma^2 + (-1 + \gamma - 1)^2 = 4 - 4\gamma + \bar{o}(\gamma)$ which is less than nearest neighbour squared distance 4 for the unmodified embedding $e$. To conclude the proof for base2 FSQ codebook, we need to consider the case when the largest perturbation with magnitude $\gamma$ corresponded to $+1$ coordinate. In this case consider $e^*$ differing from $e$ only in this coordinate and compute the squared distance $\|e' - e^*\|_2^2 = \Delta - \gamma^2 + (1 - \gamma + 1)^2 = 4 - 4\gamma + \bar{o}(\gamma)$ which is again less than 4.

Let us now consider the second base3 case. Suppose we modified the token embedding $e$ having $m$ coordinates equal to $-1$, $p$ coordinates $+1$, and $n$ zero coordinates, where $p + m + n$ is the latent space dimensionality and $p \geq 0, m \geq 0, n \geq 0$. Suppose the modified embedding $e'$ has coordinates $-1 + \delta_i$, $+1 - \beta_j$ and $\varepsilon_k$ instead of $-1$, $+1$ and $0$ for $i = 1, .., p$, $j = 1, .., m$ and $k = 1, .., n$. Perturbation values $\delta_i$ and $\beta_j$ are non-negative because we need to have $\|e'\|_\infty \leq 1$, while $\varepsilon_k$ can be either positive or negative. Denote the largest absolute value of per-coordinate perturbations by $\gamma$, i.e. $\gamma := \max\left(\max_{i=1,..,p} |\delta_i|, \max_{j=1,..,m} |\beta_j|, \max_{k=1,..,n} |\varepsilon_k|\right)$. The overall perturbation $\Delta = \sum_{i=1}^p \delta_i^2 + \sum_{j=1}^m \beta_j^2 + \sum_{k=1}^n \varepsilon_k^2$ is assumed to be sufficiently small, i.e. $\Delta = \bar{o}(\gamma)$. Perturbing a single codebook entry $e$ by modifying it to $e'$ could not increase nearest neighbour distance for any other embedding, since they all have at least two neighbours on the distance 1 (except for the trivial 1-dimensional case for which the statement of the theorem is obvious) which is the nearest neighbour distance for all token embeddings in base3 FSQ scheme. Now let us prove that the nearest neighbour distance decreases for the modified $e'$.

Suppose that the largest perturbation with magnitude $\gamma$ corresponds to $-1$ coordinate modified to $-1 + \gamma$. Consider embedding $e^*$ differing from $e$ only in this coordinate (having $0$ there instead of $-1$). Squared distance between $e'$ and $e^*$ is then calculated as $\|e' - e^*\|_2^2 = \Delta - \gamma^2 + (-1 + \gamma)^2 = 1 - 2\gamma + \bar{o}(\gamma)$ which is less than nearest neighbour squared distance 1 for the unmodified embedding $e$. Consider the case when the largest perturbation with magnitude $\gamma$ corresponded to $+1$ coordinate. In this case consider $e^*$ differing from $e$ only in this coordinate (having $0$ there instead of $+1$) and compute the squared distance $\|e' - e^*\|_2^2 = \Delta - \gamma^2 + (1 - \gamma)^2 = 1 - 2\gamma + \bar{o}(\gamma)$ which is again less than 1. Consider the final case when $\gamma$ corresponded to 0 coordinate modified to $\varepsilon$ with $\gamma = |\varepsilon|$. Take embedding $e^*$ differing from $e$ only in this coordinate and having $sign(\varepsilon)$ there instead of 0. We can compute squared distance $\|e' - e^*\|_2^2 = \Delta - \gamma^2 + (\varepsilon - sign(\varepsilon))^2 = 1 - 2\gamma + \bar{o}(\gamma)$ which concludes the proof. $\square$

# D. Proof of Best Accuracy Hypothesis in One-dimensional Case

Consider change of variables $Y_t = X_t / \alpha_t$. We know that $\alpha_0 = 1$, so $Y_0 = X_0$. Itô's formula implies that

$$dY_t = -\frac{\dot{\alpha}_t}{\alpha_t^2} X_t dt + \frac{1}{\alpha_t} dX_t = \left(-\frac{\dot{\alpha}_t}{\alpha_t^2} + \frac{f(t)}{\alpha_t}\right) X_t dt + \frac{g(t)}{\alpha_t} dW_t = \frac{g(t)}{\alpha_t} dW_t, \ \ t \in [0, T],$$

where the drift term cancels out by the formula (3). Since the SDE for $Y_t$ does not have drift term, its mean always stays the same, and the formula (2) implies that

$$Y_t = Y_0 + \frac{\sigma_t}{\alpha_t} \mathcal{N}(0, I), \ \ t \in [0, T]. \tag{23}$$

It allows us to write transitional densities of the forward processes for $X_t$ and $Y_t$ respectively:

$$p_{t|0}(x|k) = \frac{1}{(2\pi\sigma_t^2)^{n/2}} \exp\left(-\frac{1}{2\sigma_t^2} \|x - \alpha_t e_k\|_2^2\right),$$

$$p_{t|0}(y|k) = \frac{\alpha_t^n}{(2\pi\sigma_t^2)^{n/2}} \exp\left(-\frac{\alpha_t^2}{2\sigma_t^2}\|y - e_k\|_2^2\right).$$

Bayes formula implies that

$$P_{0|t}(k|x) = p_k \exp\left(-\frac{1}{2\sigma_t^2}\|x - \alpha_t e_k\|_2^2\right) / \sum_{j=1}^{V} p_j \exp\left(-\frac{1}{2\sigma_t^2}\|x - \alpha_t e_j\|_2^2\right),$$

$$P_{0|t}(k|y) = p_k \exp\left(-\frac{\alpha_t^2}{2\sigma_t^2}\|y - e_k\|_2^2\right) / \sum_{j=1}^{V} p_j \exp\left(-\frac{\alpha_t^2}{2\sigma_t^2}\|y - e_j\|_2^2\right).$$

Denote

$$\Omega_k^X(t) := \{x \in \mathbb{R}^n : k = \arg\max_j P_{0|t}(j|x)\} = \{x \in \mathbb{R}^n : k = \arg\max_j p_j \cdot \exp\left(-\frac{1}{2\sigma_t^2}\|x - \alpha_t e_j\|_2^2\right)\},$$

$$\Omega_k^Y(t) := \{y \in \mathbb{R}^n : k = \arg\max_j P_{0|t}(j|y)\} = \{y \in \mathbb{R}^n : k = \arg\max_j p_j \cdot \exp\left(-\frac{\alpha_t^2}{2\sigma_t^2}\|y - e_j\|_2^2\right)\}.$$

We consider the case $p_k = 1/V$ for all $k = 1, ..., V$, so the above definitions simplify:

$$\Omega_k^X(t) = \{x \in \mathbb{R}^n : k = \arg\min_j \|x - \alpha_t e_j\|_2^2\},$$

$$\Omega_k^Y = \{y \in \mathbb{R}^n : k = \arg\min_j \|y - e_j\|_2^2\}.$$

Note that, in contrast with the process $X_t$, the latent space area $\Omega_k^Y$ does not depend on diffusion time $t$ when we consider the process $Y_t$, this is why we prefer to manipulate with $Y_t$ in what follows.

Rewrite the average prediction accuracy $A(E, t)$ in terms of the process $Y_t$:

$$A(E, t) = \sum_{k=1}^{V} P(X_0 = e_k, X_t \in \Omega_k^X(t)) = \sum_{k=1}^{V} P(Y_0 = e_k, Y_t \in \Omega_k^Y) = \sum_{k=1}^{V} p_k P(Y_t \in \Omega_k^Y | Y_0 = e_k),$$

or, taking into account that $p_k$ are all equal,

$$A(E, t) = \frac{1}{V} \sum_{k=1}^{V} P(k = \arg\min_j \|Y_t - e_j\|_2^2 | Y_0 = e_k). \tag{24}$$

Now we will use the above formula (24) to prove the hypothesis when dimension $n = 1$. Let us first prove that if the leftmost point $e_{left} \in \mathbb{R}$ is greater than $-1$, or the rightmost point $e_{right} \in \mathbb{R}$ is less than $+1$, then moving it to its respective edge will increase $A(E, t)$. We will prove only the first case, since the second one is analogous.

Consider the leftmost point $e_{left} > -1$ and its right neighbour $e \geq e_{left}$. Let them correspond to token ids 1 and 2 respectively. It is obvious that if we move $e_{left}$ to $-1$, it will affect only probabilities of predicting token ids 1 and 2: we now predict token id 1 when $Y_t < (-1 + e)/2$ (we did it for $Y_t < (e_{left} + e)/2$ before), and we predict token 2 when $Y_t \in [(-1 + e)/2, a)$ while we did it for $Y_t \in [(e_{left} + e)/2, a)$ before ($a$ is either $+\infty$ when we consider $V = 2$, or an average of $e$ and its right neighbour when $V = 3$). As for accuracy of predicting token id 2, it increases because $P(Y_t \in [(-1 + e)/2, a)|Y_0 = e) > P(Y_t \in [(e_{left} + e)/2, a)|Y_0 = e)$ since $e_{left} > -1$. As far as accuracy of predicting token id 1 is concerned, it also increases because

$$P(Y_t < (-1 + e)/2 | Y_0 = -1) = P\left(\mathcal{N}\left(-1, \frac{\sigma_t^2}{\alpha_t^2}\right) < (-1 + e)/2\right) = P\left(\mathcal{N}\left(0, \frac{\sigma_t^2}{\alpha_t^2}\right) < (1 + e)/2\right) >$$

$$> P\left(\mathcal{N}\left(0, \frac{\sigma_t^2}{\alpha_t^2}\right) < (-e_{left} + e)/2\right) = P\left(\mathcal{N}\left(e_{left}, \frac{\sigma_t^2}{\alpha_t^2}\right) < (e_{left} + e)/2\right) = P(Y_t < (e_{left} + e)/2 | Y_0 = e_{left}).$$

We used conditional distribution (23) of $Y_t | Y_0$ and the fact that $e_{left} > -1$ to derive the above inequality.

Thus, we have proven that the leftmost and the rightmost points should be $-1$ and $+1$ respectively. It concludes the proof for base2 FSQ codebook. To finish the proof of the hypothesis in base3 case, suppose that the middle point equals $\alpha$ and maximize $A(E(\alpha), t)$ for $E = \{-1, \alpha, +1\}$.

Denote cumulative distribution function of standard normal random variable $\xi$ by $\Phi(x)$ and its probability density function by $\varphi(x)$. Calculate each term in the expression (24) for average prediction accuracy:

$$3 \cdot A(E(\alpha), t) = P\left(-1 + \frac{\sigma_t}{\alpha_t}\xi \leq \frac{-1+\alpha}{2}\right) + P\left(\frac{-1+\alpha}{2} \leq \alpha + \frac{\sigma_t}{\alpha_t}\xi \leq \frac{1+\alpha}{2}\right) + P\left(1 + \frac{\sigma_t}{\alpha_t}\xi \geq \frac{1+\alpha}{2}\right) =$$

$$= P\left(\xi \leq \frac{\alpha_t}{\sigma_t}\frac{1+\alpha}{2}\right) + P\left(-\frac{\alpha_t}{\sigma_t}\frac{1+\alpha}{2} \leq \xi \leq \frac{\alpha_t}{\sigma_t}\frac{1-\alpha}{2}\right) + P\left(\xi \geq -\frac{\alpha_t}{\sigma_t}\frac{1-\alpha}{2}\right) =$$

$$= \Phi\left(\frac{\alpha_t}{\sigma_t}\frac{1+\alpha}{2}\right) + \Phi\left(\frac{\alpha_t}{\sigma_t}\frac{1-\alpha}{2}\right) - \Phi\left(-\frac{\alpha_t}{\sigma_t}\frac{1+\alpha}{2}\right) + 1 - \Phi\left(-\frac{\alpha_t}{\sigma_t}\frac{1-\alpha}{2}\right) =$$

$$= 2 \cdot \left(\Phi\left(\frac{\alpha_t}{\sigma_t}\frac{1+\alpha}{2}\right) + \Phi\left(\frac{\alpha_t}{\sigma_t}\frac{1-\alpha}{2}\right)\right) - 1$$

In the above derivation we used symmetry of standard normal distribution, namely the fact that $\Phi(x) + \Phi(-x) = 1$. Now compute the first and second derivative of $A(E(\alpha), t)$ with respect to $\alpha$:

$$A'(E(\alpha), t) = \frac{1}{3}\frac{\alpha_t}{\sigma_t}\left(\varphi\left(\frac{\alpha_t}{\sigma_t}\frac{1+\alpha}{2}\right) - \varphi\left(\frac{\alpha_t}{\sigma_t}\frac{1-\alpha}{2}\right)\right)$$

$$A''(E(\alpha), t) = \frac{1}{6}\frac{\alpha_t^2}{\sigma_t^2}\left(\varphi'\left(\frac{\alpha_t}{\sigma_t}\frac{1+\alpha}{2}\right) + \varphi'\left(\frac{\alpha_t}{\sigma_t}\frac{1-\alpha}{2}\right)\right)$$

Since we search for $\alpha \in [-1, 1]$, then the only solution to the equation $A'(E(\alpha), t) = 0$ is $\alpha = 0$, and it is easy to see that for $\alpha = 0$ $A''(E(\alpha), t) < 0$ because both $\alpha_t$ and $\sigma_t$ are positive. So, average prediction accuracy $A(E, t)$ is maximized when $E = E_{FSQ} = \{-1, 0, +1\}$ for all $t$.

## E. Architecture of CDCD-TTS

CDCD-TTS model differs from CosyVoice2 only by a module generating speech tokens from text and a duration predictor. Below one can find a detailed description of these modules.

Duration predictor in CDCD-TTS estimates the number of speech tokens corresponding to input text from statistics collected from sufficiently large annotated speech corpus. A simple model assigning each text character its average duration expressed in the number of speech tokens is fit to this corpus by means of minimizing MSE loss. At CDCD-TTS inference, this statistical model is first applied to reference text, and the scalar $\kappa$ is calculated as the actual number of tokens corresponding to the reference audio divided by the number of speech tokens predicted by the statistical model. The value of $\kappa$ shows the relative speed of the reference speech. The number of speech tokens corresponding to the input text is then computed as the number of tokens predicted by the statistical model from the input text multiplied by $\kappa$.

CDCD backbone borrows its architecture from F5-TTS. In contrast with CosyVoice2 model utilizing BPE-based text tokenizer, CDCD-TTS model recognizes input English text just as a sequence of latin characters, digits and punctuation marks. The input text conditioning, i.e. reference and input texts padded to the same length as the sum of the number of reference tokens and the number of tokens predicted by the duration predictor, is passed through 4 ConvNeXt v2 blocks with hidden dimension 256. The output is concatenated with the reference clean latent vectors and input noisy latent vectors to form input to a stack of 8 DiT (adaLN-zero) blocks each having 8 attention heads with the inner dimension 512 and RoPE embeddings. DiT blocks are followed by the softmax layer predicting probabilities of 6561 classes, i.e. FSQ speech tokens. Diffusion time $t$ is encoded with sinusoidal positional embedding and DiT context length is 1024, i.e. the number of reference and generated speech tokens should be no more than 1024, or approximately 41 seconds.

