# OpenReview forum: "Optimality of FSQ Tokens for Continuous Diffusion for Categorical Data with Application to Text-to-Speech"
_ICML.cc/2026/Conference — ICML 2026 regular_

### Official Review · Reviewer_QCio · 2026-03-04

**Soundness:** 3
**Presentation:** 4
**Significance:** 3
**Originality:** 3
**Overall Recommendation:** 4
**Confidence:** 3

**Summary:**

This paper studies the latent-space geometry of Continuous Diffusion for Categorical Data (CDCD) and argues that Finite Scalar Quantization (FSQ) provides a particularly suitable codebook structure. The paper gives theoretical justification through a link between diffusion-path KL divergence and distances between token embeddings, formulates a “Best Accuracy Hypothesis” for token prediction, and validates these ideas both analytically and empirically.

**Compliance With Llm Reviewing Policy:**

Affirmed.

**Final Justification:**

I kept my score.

**Key Questions For Authors:**

1. Regarding the "equal prior probability" assumption: have you tested the prediction accuracy on a non-uniform distribution? Does the FSQ structure still hold its advantage there?
2. In Table 1, the gap between FSQ-original and FSQ-perturb narrows as the number of diffusion steps increases. Does this imply that with enough compute, the latent space structure becomes irrelevant, or are there fundamental limits that the perturbed version can never overcome?
3. Could you clarify the choice of the $L_\infty$ constraint? If we allow the codebook to be constrained by a fixed total energy ($L_2$ norm), would a spherical arrangement be superior to the FSQ hypercube?
4. How does the model handle very long or very short utterances where the "average character duration" might be an inaccurate heuristic?

**Limitations:**

See Weaknesses and Questions.

**Strengths And Weaknesses:**

**Strengths**
- The work is a complete "theory-to-application" pipeline, which is rare. It starts with SDE proofs and ends with a SOTA-beating TTS system.
- The performance gains are impressive: a 10x reduction in parameter count with a concurrent increase in quality is a strong signal of methodological correctness.
- The paper is well-written and exceptionally clear in its notation and logical flow, making complex stochastic calculus concepts accessible.

**Weaknesses**
1. Theorem 4.3 only proves local optimality. While the numerical experiments in Section 5.1 suggest global optimality, a formal proof for the global case is missing.
2. As noted, the accuracy hypothesis assumes $p_k = 1/V$. In categorical data like text or speech tokens, the distribution is highly long-tailed. The paper does not theoretically address how FSQ performs when some tokens are orders of magnitude more frequent than others.
3. The "simple statistical duration predictor" mentioned in Section 5.2 feels like an engineering workaround compared to the principled diffusion approach used elsewhere. If the duration prediction fails, the CDCD model's performance is bottlenecked regardless of latent space optimality.
4. In Figure 2, the "discarded" box is not explicitly explained.
5. The authors should consider providing an ablation study on the impact of the duration predictor. For instance, how does the model perform if "ground truth" durations are used versus the statistical predictor? This would isolate the strength of the CDCD module.

---

### Official Review · Reviewer_dKT4 · 2026-03-11

**Soundness:** 3
**Presentation:** 2
**Significance:** 2
**Originality:** 3
**Overall Recommendation:** 4
**Confidence:** 4

**Summary:**

This paper studies the influence of the latent space structure corresponding to discrete tokens on the training of continuous diffusion models. The authors analyze the geometric properties of the latent space from the perspective of the Kullback–Leibler divergence between diffusion path measures, and establish a theoretical connection between reverse diffusion path measures and the distance between generated samples along these trajectories.
Based on this theoretical framework, the paper analyzes the latent space properties of FSQ (Finite Scalar Quantization) tokenization and argues that: The latent space geometry produced by FSQ is particularly suitable for Continuous Diffusion for Categorical Data (CDCD). Under the assumption of an optimally trained diffusion model, the FSQ latent space may maximize the average token prediction accuracy. On the application side, the authors further develop a CDCD-based text-to-speech (TTS) system and demonstrate its performance on zero-shot voice cloning. The results suggest superior generation quality compared to methods based on large language models (LLMs).

**Compliance With Llm Reviewing Policy:**

Affirmed.

**Final Justification:**

I decide to maintain my current score without change.

**Key Questions For Authors:**

1.How can it be empirically verified that the latent geometry of FSQ indeed improves diffusion performance?
2.The theoretical results assume an optimally trained diffusion model. Do the conclusions still hold under suboptimal training conditions, which are more common in practice?
3.Could the performance improvements be explained by architecture differences or training tricks, rather than the latent geometry itself?

**Limitations:**

The paper provides an interesting theoretical perspective on the relationship between latent space geometry and diffusion training for categorical data, and offers an analysis of why FSQ tokenization may be particularly suitable. While the theoretical insights are valuable, the methodological novelty is somewhat limited and the empirical validation could be stronger to fully support the claims.

**Strengths And Weaknesses:**

Strengths
1. Novel theoretical perspective
The paper attempts to answer a relatively underexplored question: Does the geometric structure of tokenization latent spaces influence diffusion training?
By introducing an analysis framework based on KL divergence between diffusion trajectories, the paper provides a geometric interpretation of token embeddings. This represents a relatively novel theoretical perspective.

2. Establishing a theoretical link between tokenization and diffusion
The paper proposes a conceptual chain: token embedding geometry → diffusion trajectory distance → prediction accuracy
This connection is conceptually meaningful. In particular, linking reverse diffusion path measures with latent space distances, and interpreting the advantages of FSQ codebook geometry from an information-theoretic perspective, is not commonly explored in the current diffusion literature.

3. Complete structure with theory and applications
The paper presents a relatively complete structure, including: theoretical analysis, numerical experiments, an implemented system (TTS).
In particular, the implementation of a CDCD-based TTS system demonstrates that the proposed insights are not purely theoretical but also have practical and engineering potential. The results therefore suggest possible real-world applicability.

Weaknesses
1. The contribution is primarily analytical rather than methodological
The core contribution of the paper mainly lies in analyzing the latent geometry of FSQ and explaining why it is suitable for diffusion. However, the work does not propose a new tokenization method, or a new diffusion framework.
Therefore, the contribution may be viewed more as theoretical explanation rather than a new algorithmic advancement.

2. FSQ itself is not a new technique
FSQ (Finite Scalar Quantization) is not introduced in this paper. As a result, the contribution mainly lies in:theoretical interpretation + analysis of diffusion suitability, rather than proposing a new tokenization mechanism.

3. The TTS experiment may not fully validate the theoretical claims
The theoretical claim of the paper concerns the relationship between latent geometry → diffusion performance, However, the experimental evaluation mainly focuses on TTS quality.
It remains unclear whether the observed performance improvements are directly caused by the latent geometry, rather than other factors.

---

### Official Review · Reviewer_u4Vf · 2026-03-11

**Soundness:** 2
**Presentation:** 3
**Significance:** 2
**Originality:** 3
**Overall Recommendation:** 3
**Confidence:** 4

**Summary:**

Paper proposes a theoretical analysis and proves FSQ tokens are optimal for representing categorical variables under continuous diffusion processes, followed by experiments to show the efficiency in TTS application.

**Compliance With Llm Reviewing Policy:**

Affirmed.

**Key Questions For Authors:**

Given that successful TTS systems employ RVQ and Diffusion models, the performance of “RVQ-25” in Table 1—such as its WER—is unexpectedly poor. Is there a specific reason that accounts for this anomaly?

**Limitations:**

In addition to the statement of "Optimality Properties of FSQ," it's better to specify the mathematical assumptions behind this claim. And these assumptions might not always apply in real-world scenarios.

**Strengths And Weaknesses:**

Strength: The paper provides a thorough theoretical analysis and proves the “Optimality Properties of FSQ” within the CDCD model.

Weakness: There is a lack of experimental comparisons with stronger baselines in TTS. Additionally, some results in experiments (e.g. on RVQ) are not sufficiently convincing.

---

### Official Review · Reviewer_vtZh · 2026-03-13

**Soundness:** 3
**Presentation:** 3
**Significance:** 2
**Originality:** 3
**Overall Recommendation:** 3
**Confidence:** 3

**Summary:**

This paper studies the properties of the optimal structure of the latent token space for CDCD models, which, in particular, addresses the “rounding” problem (discretization error) in CDCD training. Specifically, the authors first showed in theory that, in general continuous diffusion, the identifiability of two reverse diffusion paths is a function of the $L^2$ distance between the target termination points. Second, they showed that FSQ (specifically base2 and base3) preserves local optimality, i.e., achieving the minimal average closest neighbor distance (an aggregation over of $L^2$ distances to neighbors) locally, which in turn proves that FSQ codebook is well-suited for CDCD models. Finally, the authors experimented with theoretical findings in the task of speech synthesis, achieving strong performance compared to the baseline model.

**Compliance With Llm Reviewing Policy:**

Affirmed.

**Final Justification:**

Keeping my score as no rebuttal from the authors.

**Key Questions For Authors:**

1. In Line#428, what do the steps mean? You mean the denoising steps?

2. What is the tokenization used for the LLM-based models?

3. The effect of the duration predictor is unclear. Can you provide a more detailed ablation analysis for this?

4. Minor typos: e.g., Line#689 should be a_1 in the equation.

**Limitations:**

Please see my comments above.

**Strengths And Weaknesses:**

**Strengths**

1. The paper is very well motivated and clearly written.

2. The insight of improving CDCD models from the angle of the compatibility of tokenization is novel and very interesting.

3. The results in TTS are strong.

**Weaknesses**

1. The theoretical contributions are bloated imho. Theorem 4.1 is a direct derivative of the Girsanov theorem, and shouldn't be a theorem by itself. The core contribution is the local optimality of the FSQ codebook in terms of the average nearest neighbor distances. However, the global optimality is only numerically illustrated with low dimensionality cases. A theoretical proof will make the paper stronger.

2. Lack of comparison against other tokenization algorithms, specifically, Lookup-Free Quantization (LFQ), VQ-VAE.

3. In the case of TTS, the size of the vocabulary generated with FSQ is only 6561, which is relatively simple and small. It’s unclear how this method can be scaled up to larger domains with more complex vocabulary, which limits the impact and significance of this research.

4. The duration predictor module seems critical. However, it lacks sufficient description of the design details, nor is it ablated. For example, how about sampling the length/duration from a fixed prior distribution?

5. Lack of sufficient detail for reproduction, specifically, the author should at least provide the tables of the hyperparameters used in the TTS experiments in this paper.

6. While the authors' attempt to create a self-contained paper is commendable and enhances readability, allocating four pages of the main text to background setup may be a minor drawback, as it potentially diminishes the space available for highlighting the work's technical novelty.

---

### Decision · Program_Chairs · 2026-04-30

**Decision:**

Accept (regular)

**Comment:**

This paper studies the properties latent token space for Continuous Diffusion for Categorical Data (CDCD) models. Theoretically, the authors show FSQ codebooks are especially suitable for CDCD. Empirically, strong performance compared to the baseline model is demonstrated through a text-to-speech application.

### Strengths

- The paper addresses an interesting and relatively underexplored question: whether tokenization geometry itself affects continuous diffusion for categorical data.
- The theory-to-application structure is a real strength.
- The core perspective is novel. In particular, relating latent geometry to diffusion-path KL divergence and then to token prediction behavior offers a useful lens on CDCD models.
- The TTS results are promising.

### Weaknesses

- The main theoretical claim is not fully complete. The paper proves local optimality of FSQ codebooks, but global optimality is only supported numerically in low-dimensional settings.
- The empirical validation does not fully isolate latent geometry as the factor behind the TTS gains. Several reviewers noted that other system choices, especially the duration predictor and the full TTS stack, could also affect the outcome.
- The experimental comparisons are not as strong. Reviewers asked for stronger tokenization baselines, as well as stronger TTS baselines.

### Rebuttal and Remaining Concerns

The authors submitted late confidential comments to the AC addressing reviewer concerns. Those comments are useful and partially strengthen the paper. From the AC's perspective, the main remaining concerns are:

- the gap between local theoretical optimality and the broader claims suggested by the application results;
- the limited empirical isolation of latent geometry from other modeling choices;
- the lack of stronger baseline comparisons; and
- the scalability of the approach to larger and more complex vocabularies.

Therefore, this is a borderline paper. It has strengths and could be viewed as a weak accept.